# Clinical Insights into Non-Alcoholic Fatty Liver Disease and the Therapeutic Potential of Flavonoids: An Update

**DOI:** 10.3390/nu17060956

**Published:** 2025-03-09

**Authors:** Aleksandra Kozłowska

**Affiliations:** Department of Social Medicine and Public Health, Medical University of Warsaw, 02-106 Warsaw, Poland; aleksandra.kozlowska@wum.edu.pl

**Keywords:** NAFLD, flavonoids, liver steatosis, quercetin, liver enzymes

## Abstract

Non-alcoholic fatty liver disease (NAFLD) is considered a significant global health issue related to serious metabolic disorders. However, effective pharmacological treatments are still limited. Flavonoids, a wide group of polyphenol substances, exert anti-inflammatory and lipid-lowering effects in preclinical data. Thus, implementing these research findings in clinical practice could significantly help manage NAFLD and its consequences. This narrative review assesses the therapeutic potential of flavonoids in managing NAFLD. The research collected randomized controlled trials (RCTs) and meta-analyses of RCTs from the past five years concerning the impact of flavonoids on NAFLD. A total of 20 studies were selected according to predetermined inclusion criteria, comprising thirteen randomized controlled trials (RCTs) and seven meta-analyses. The research underscores the beneficial effects of flavonoids in the management of NAFLD through the enhancement of lipid metabolism, the reduction in hepatic steatosis, and the provision of anti-inflammatory actions. Clinical trials demonstrate that interventions rich in flavonoids, including quercetin, epigallocatechin gallate, naringenin, and isoflavones, substantially reduce liver fat content and enhance liver enzyme profiles, with certain compounds exhibiting superior efficacy in particular subgroups, such as older adults and females. Nonetheless, whereas these therapies significantly diminish hepatic steatosis, their effect on fibrosis is constrained. To sum up, flavonoids exhibit significant potential as supplementary treatments for NAFLD by enhancing liver function, lipid metabolism, and inflammation. Additional extensive controlled clinical trials are necessary to create uniform treatment methods and ascertain their long-term therapeutic advantages.

## 1. Introduction

Non-alcoholic fatty liver disease (NAFLD) has been recognized as a major worldwide health issue [1]. The primary cause of NAFLD is the excessive accumulation of fat within hepatocytes, which account for over 5% of the liver’s weight or volume. Approximately 32% of adults are affected by NAFLD, with a higher prevalence among males than females, based on the present estimates [2,3]. Recently, a new classification termed metabolic-associated fatty liver disease (MAFLD) was introduced [4]. This condition is diagnosed when liver steatosis is present alongside at least one criterion related to metabolic syndrome. While MAFLD refers to the early stages of liver disease characterized by a fatty liver without significant inflammation, it can progress to more advanced stages, such as metabolic-associated steatohepatitis (MASH). MASH, which was previously referred to as NASH (non-alcoholic steatohepatitis), is characterized by a more severe accumulation of fat and liver inflammation [5]. This condition can result in hepatocellular injury and, in the long term, liver fibrosis, cirrhosis, and an elevated risk of liver cancer [6,7].

NAFLD is mainly caused by overnutrition, which results in the accumulation of visceral fat. In this context, the infiltration of macrophages into the visceral adipose tissue compartment induces a pro-inflammatory state that leads to insulin resistance. Thus, it leads to an increased influx of free fatty acids into the liver, enhanced de novo lipogenesis, and impaired fatty acid oxidation [1]. At the same time, the dysregulation of lipid metabolism results in the accumulation of lipotoxic lipids, which induce cellular stress, activate inflammatory pathways, and cause hepatocyte damage. The pathogenic mechanisms of NAFLD are also affected by several metabolic, genetic, and microbiome-related variables. However, they remain not fully understood [1].

Despite the high prevalence of MAFLD and NASH, effective treatment options remain limited. Lifestyle interventions, including a restricted-calorie weight-loss diet and regular exercise, are the foundational strategies for managing NAFLD [8,9]. Nevertheless, the absence of approved medical treatments has caused an interest in therapeutic alternatives, such as dietary supplements. Among patients with NAFLD, compounds such as flavonoids have shown potential to enhance liver function [10,11,12].

Flavonoids are a group of phytochemicals present in vegetables and fruits. Key flavonoid groups include flavonols, flavones, flavanones, flavanols, anthocyanins, and isoflavones [13,14]. These substances, which are recognized for their antioxidant, anti-inflammatory, and lipid-lowering properties, have been proposed to provide substantial metabolic health benefits to patients with hepatic steatosis. However, despite their promising biological activities, flavonoids frequently demonstrate relatively low effectiveness, which frequently requires high doses of these substances to obtain therapeutic effects. Additionally, the poor bioavailability of many flavonoids presents significant challenges for their clinical translation [15]. Strategies such as structural modifications, nanoformulations, and co-administration with enhancers are being explored to overcome these limitations [16].

Nevertheless, the rising interest in the potential therapeutic applications of flavonoids is evidenced by the increasing amount of clinical research assessing their efficacy and safety across diverse health conditions. In a query conducted in the ClinicalTrials.gov (accessed on 20 February 2025) database in February 2025, the keyword “flavonoid” yielded 179 clinical studies in which these compounds were investigated. The current study aims to clarify the relationship between flavonoid consumption, whether through dietary supplements, pure compounds, combinations, or extracts, and its advantageous effects in patients with NAFLD.

## 2. Materials and Methods

The author conducted a literature search using the PubMed database to identify relevant studies on the relationship between flavonoids, specific flavonoid subclasses, non-alcoholic fatty liver disease, non-alcoholic steatohepatitis, and liver steatosis. A search was conducted on PubMed with the following terms in titles and abstracts: (flavonoids OR “flavonoid-rich” OR anthocyanins OR flavonols OR flavones OR flavanones OR flavanols OR isoflavones OR quercetin OR kaempferol OR myricetin OR isorhamnetin OR luteolin OR apigenin OR hesperidin OR naringenin OR eriodictyol OR catechin OR epigallocatechin OR cyanidin OR delphinidin OR malvidin OR pelargonidin OR peonidin OR petunidin OR silymarin) AND (NAFLD OR ”non-alcoholic fatty liver disease ” OR MAFLD OR “metabolic-associated fatty liver disease” OR MASH OR “metabolic-associated steatohepatitis” OR NASH OR “non-alcoholic steatohepatitis” OR “liver steatosis” OR “liver fibrosis” OR “liver cirrhosis”). The search was performed on 20 January 2025 and included studies published between 1 January 2020 and 20 January 2025.

For study selection, the author applied the following inclusion criteria: (1) randomized controlled trials (RCTs); (2) clinical trials; (3) meta-analyses; (4) studies published in English; and (5) studies published in the last 5 years. The exclusion criteria were as follows: (1) studies not available in English, (2) case reports, (3) narrative reviews, and (4) studies focusing on nonclinical (in vitro or animal model) outcomes. After screening the titles and abstracts, studies that did not meet the inclusion criteria or contained misleading data were excluded, resulting in 20 eligible studies (13 RCTs and 7 meta-analyses). The selection process involved assessing the study relevance based on the predefined inclusion and exclusion criteria, followed by full-text screening. Data extraction was performed using EndNote X7, focusing on the study design, population characteristics, intervention details, flavonoid type, dosage, duration, and primary clinical outcomes (Appendix A). Preclinical data were not limited, regardless of their publication by publication date. However, the author has included a review of the most recent studies available up to the current date that the readers may not be familiar with.

## 3. Dietary Sources, Intake, and Safety of Flavonoids

Epidemiological studies indicate significant differences in flavonoid intake among populations. For example, in Australia, the daily intake of total flavonoids was 626 ± 579 mg, while in European countries, the mean intake was estimated at 428 ± 49 mg/day [17,18]. A smaller intake of flavonoids has been reported in United States adults, where the mean daily intake was 344.83 ± 9.13 mg [19]. Nevertheless, the results of nutritional assessments of flavonoid intake are difficult to estimate due to the various data sources and analytical methods used. It is crucial to take the publication dates of the data into account when evaluating flavonoid intake across different populations, as previously, databases on the content of specific flavonoids were incomplete. Currently, the US Department of Agriculture (USDA) and Phenol-Explorer databases are the most widely used databases for studying the flavonoid content of foods [20].

The flavonoid content in food varies based on the horticultural crop variety, the cultivation method, and the specific part of the plant [21]. Table 1 presents data regarding the flavonoid contents of particular food categories, including fruits, vegetables, spices and herbs, nuts, beverages, legumes, sweets, cereals, and grains. The highest content of flavonoids is found in spices and herbs (4854.49 mg/100 g in dried parsley and 1545.79 mg/100 g in dried Mexican Oregano), in fruits, including berries (518.05 mg/100 g in elderberries and 368.66 mg/100 g in chokeberries), and in beverages (118.35 mg/100 mL in black tea, 53.84 mg/100 mL in green tea) [22,23].

The consumption of dietary flavonoids is considered safe when derived from natural products [24]. However, the increasing numbers of pharmaceutical supplements containing concentrated flavonoids have raised several concerns [25,26]. The dearth of studies on the safety of these compounds exacerbates this concern. Flavonoids may have unknown adverse effects or interact poorly with medications and other food components [27]. Furthermore, high dosages of specific flavonoids may interfere with absorbing nutrients such as trace elements and vitamins, especially folic acid. Additionally, there have been reports of toxic flavonoid–drug interactions, liver failure, contact dermatitis, hemolytic anemia, and estrogenic-related issues including male reproductive health and breast cancer that have been linked to exposure to dietary flavonoids or phenolics. [28,29,30].

## 4. Flavonoids and Their Actions in NAFLD

Flavonoids and their metabolites demonstrate a variety of biochemical properties. Current studies regarding their potential as a therapeutic agent for NAFLD focus on findings investigating the cellular and molecular mechanisms involved. Recent preclinical studies, including both in vitro and in vivo models, provide evidence of flavonoids’ favorable impact on lipid regulation, steatosis improvement, and inflammation reduction under NAFLD conditions (Table 2) [31,32,33,34,35,36,37,38].

Carvalho et al. investigated the impact of silymarin intake on hepatic lipid dysfunction in NAFLD mice, where NAFLD was induced through a 30% fructose diet [39]. The silymarin treatment improved liver function by decreasing the levels of aspartate aminotransferase (AST), alanine aminotransferase (ALT), superoxide dismutase/catalase (SOD/CAT), thiobarbituric acid reactive substances (TBARSs), hypertriglyceridemia, and hypercholesterolemia. Furthermore, silymarin increased the mRNA levels and activity of hepatic citrate synthase, suggesting that silymarin may reduce the worsening of NAFLD in the animal model [39]. Similarly, other flavonoids, such as isorhamnetin, exhibited lipid-lowering effects in fatty acid-induced HepG2 and BEL-7402 cell lines and high-fat diet mice [40]. Furthermore, isorhamnetin modulated bile acid metabolism via farnesoid X receptor (FXR) and bile salt export pump (BSEP) upregulation, suggesting that this flavonoid may improve hepatic bile acid homeostasis.

Quercetin, one of the most abundant flavonols, enhanced lipid metabolism in C57BL/6 mice fed a high-fat diet by reducing lipogenesis-related proteins such as acetyl-CoA carboxylase (ACC), fatty acid synthase (FASN), and sterol regulatory element-binding protein 1c (SREBP-1c). Additionally, these substances promoted β-oxidation by upregulating peroxisome proliferator-activated receptor alpha (PPARα) and carnitine palmitoyltransferase 1A (CPT1A) [41]. Interestingly, quercetin enhanced the antioxidant capacity in high-fat diet-treated mice by downregulating nuclear factor erythroid-2-related factor 2 (Nrf2) and heme oxygenase-1 (HO-1) expression and upregulating SOD and glutathione peroxidase 1 (GPX1) expression. These findings suggest that quercetin improves lipid metabolism and antioxidant capacity in high-fat diet-induced metabolic dysfunction by reducing lipogenesis, promoting β-oxidation, and modulating oxidative stress-related pathways.

In a model of non-alcoholic steatohepatitis in female mice, daily oral administration of 20 mg/kg kaempferol for 12 weeks modulated systemic inflammatory responses and lipid metabolism [42]. The kaempferol treatment reduced the levels of total cholesterol (TC), triglycerides (TGs), and low-density lipoprotein cholesterol (LDL-c) and improved high-density lipoprotein cholesterol (HDL-c) in NASH mice models. Furthermore, a decrease in inflammatory factors and a modulation of the NOD-, LRR-, and pyrin domain-containing protein 3/target of methylation-induced silencing/Caspase-3 (NLRP3-ASC/TMS1/Caspase-3) pathway were observed. The NLRP3-ASC/TMS1/Caspase-3 pathway is a critical component of inflammation and initiated liver injury in NASH mice. Therefore, the protective effects of kaempferol against MASH could be mediated by mitigating liver damage and inflammation by targeting NLRP3 [42].

Another promising flavonoid, 7-Hydroxyflavone (7-HY), was demonstrated to diminish fat accumulation, hepatic steatosis, and oxidative stress in high-fat diet mice. Subsequent experimental results showed 7-HY’s ability to reduce TG deposition in HepG2 cells via action with serine/threonine kinase 24 (STK24), a pathway that affects the lipid metabolism and inflammatory processes. This suggests that 7-HY may attenuate the inflammatory processes associated with NAFLD progression [43]. Moreover, *Broussonetia papyrifera* flavonoids have been demonstrated to initiate the Nrf2 pathway, leading to a reduction in reactive oxygen species (ROS) and an increase in antioxidant enzyme activity, such as SOD [44]. This mechanism contributes to the alleviation of oxidative stress and inflammation in NAFLD models.

Flavonoids exhibit significant potential in modulating lipid metabolism, reducing hepatic steatosis, and mitigating inflammation in NAFLD. Preclinical studies have demonstrated their ability to regulate key metabolic pathways, decrease lipogenesis, and enhance β-oxidation, improving lipid homeostasis. Additionally, these substances exert anti-inflammatory effects by modulating both the NLRP3 and STK24 pathways, thereby reducing oxidative stress and liver damage. These results highlight the therapeutic potential of flavonoids in preventing NAFLD and the progression of NAFLD to more severe types of liver disorders.

**Table 2 nutrients-17-00956-t002:** An update on in vivo and in vitro studies (research published in 2024) regarding flavonoids and their prevention and treatment effects in NAFLD.

Disorder/Substances	In Vitro or In Vivo Model		Mode of Action	References
Quercetin	MAFLD C57BL/6 mice fed a high-fat diet		improvement of lipid metabolism by reducing lipogenesis proteins, including ACC, FASN, and SREBP-1c, and enhancing β-oxidation proteins, including PPARα and CPT1A, enhanced antioxidant capacity, eliminating phosphorylation of IκBα and NF-κB p65	Jiang et al. [41]
Silymarin	mice fed with 30% fructose	↓ ↓ ↓ ↑	AST, ALT, SOD/CAT, TBARSs, TGs, TC, ACC-1, and FAS mRNA levels and activity of hepatic citrate synthase	Carvalho et al. [39]
Isorhamnetin	HepG2 and BEL-7402 cell lines induced with FFAs	↓ ↓	intracellular lipid deposition TGs, TC upregulating FXR and BSEP and downregulating SLCO1B3	La et al. [40]
Purification of flavonoids from quinoa whole grain	HepG2 and BEL-7402 fatty liver cell models induced by OA and PA in a high-fat diet-induced NAFLD model in C57BL/6N mice	↓ ↓ ↓ ↑	TGs (in both models) TC, LDL-C in mice AST, ALT in mice HDL-C in mice downregulating lipid metabolism genes, CD36, and FASN	La et al. [31]
Erhuang Quzhi Formula rich in luteolin and quercetin	WRL68 or HepG2 cells induced by OA and PA; male ICR mice	↓	lipid accumulation regulated changes in adipose tissue, inhibited MAPK/AKT signaling pathway	Pan et al. [32]
7-Hydroxyflavone (7-HY)	oleic acid/palmitic acid-induced HepG2 cells and C57BL/6 mice on a high-fat diet		mitigated fat accumulation, hepatic steatosis, and oxidative stress induced by high-fat diet ameliorating abnormal glucose metabolism inhibited TG deposition in HepG2 cells through interaction with STK24	Qi et al. [43]
Grape Seed Proanthocyanidin Extract (GSPE)	CAF-induced liver steatosis male Fischer 344 rats	↓	hepatic triglyceride levels modulated liver physiological processes	Rodriguez et al. [33]
Vine tea (*Ampelopsis grossedentata*)	mice fed a high-fat diet; HepG2 and L02 cells treated with OA	↓ ↓	body mass blood lipids improved hepatic tissue morphology activation of AMPK/mTOR	Wang et al. [34]
*Broussonetia papyrifera*	HepG2 cells treated with FFAs; mice fed with a high-fat diet	↓ ↑ ↑	TC, LDL-C, TGs HDL-C inhibited generation of ROS; restrained level of MPO, activity of superoxide SOD activation of Nrf2	Wang et al. [44]
Quercetin and kaempferol from *Carthamus* *tinctorius* L.	a rat model of NAFLD; HepG2 cells		enhancing bile acid receptor NR1H4 expression	Wang et al. [35]
Kaempferol	C57BL/6 female mice on a high-fat diet	↓	expression of NLRP3- ASC/TMS1-Caspase 3 downregulated expression of NLRP3-ASC/TMS1-Caspase 3	Yang et al. [42]
Dihydromyricetin	PA-treated HepG2 cells; rats fed with a high-fat diet	↑ ↓	ameliorating hepatic steatosis, GLUT2 expression, and G6Pase and PEPCK expression improved IR	Yang et al. [36]
Chrysin	NAFLD model cells; db/db mice	↓ ↓ ↓ ↓	hyperlipidemia liver injury body weight liver weight	Zhang et al. [37]
Apigenin-6-C-glucoside	HepG2 and PLC/PRF/5 liver cell lines;murine models of MASLD	↓ ↓ ↑	hepatic steatosis and fibrosis, pro-inflammatory macrophage numbers hepatic glycogen content	Khatoon et al. [38]

↑—increase, ↓—decrease, ACC—acetyl-CoA carboxylase, ALT—alanine aminotransferase, AMPK—AMP-activated protein kinase, ASC—apoptosis-associated speck-like protein containing a CARD, AST—aspartate aminotransferase, BSEP—bile salt export pump, CAF—cancer-associated fibroblast, CAT—catalase, CD36—cluster of differentiation 36, FASN—fatty acid synthase, HDL-c—high-density lipoprotein cholesterol, FFAs—free fatty acids, FXR—farnesoid X receptor, G6Pase—glucose-6-phosphatase, GLUT2—glucose transporter 2, IκBα—inhibitor of nuclear factor kappa B alpha, IR— insulin receptor, LDL-c—low-density lipoprotein cholesterol, MAPK—mitogen-activated protein kinase, MASLD—metabolic dysfunction-associated steatotic liver disease, MPO—metalloproteinase, mTOR—mechanistic target of rapamycin, NAFLD—non-alcoholic fatty liver disease, NF-κB p65—nuclear factor kappa B p65, NLRP3—NOD-, LRR-, and pyrin domain-containing protein 3, NR1H4—nuclear receptor subfamily 1 group H member 4, Nrf2—nuclear factor erythroid-2-related factor 2, OA—oleic acid, PA—palmitic acid, PEPCK—phosphoenolpyruvate carboxykinase, PPARα—peroxisome proliferator-activated receptor alpha, ROS—reactive oxygen species, SLCO1B3—solute carrier organic anion transporter family member 1B3, SOD—superoxide dismutase, SREBP-1c—sterol regulatory element-binding protein 1c, STK24—serine/threonine kinase 24, TBARS—thiobarbituric acid reactive substances, TMS1—target of methylation-induced silencing 1, TC—total cholesterol, TG—triglyceride.

## 5. Results

Flavonoids have been extensively supported by preliminary research as substances with hepatoprotective effects in NAFLD by reducing liver fat accumulation, oxidative stress, and inflammation through modulating key metabolic pathways. However, the evidence in humans remains limited. In recent years, numerous randomized controlled trials and meta-analyses have been published (Appendix A). This update highlights the results from clinical studies that offer the most compelling evidence for the impact of flavonoid intake on improving metabolic status, reducing inflammation, and modulating lipid profiles in patients with NAFLD.

### 5.1. Flavonoids and Liver-Related Changes in NAFLD

Liver steatosis, defined by excessive fat deposition in hepatocytes, is an important hallmark of NAFLD and a primary factor in the disease progression [45,46]. Chronic hepatic lipid accumulation can induce inflammation and fibrotic remodeling, increasing the risk of non-alcoholic steatohepatitis, cirrhosis, and hepatocellular cancer. Advanced fibrosis, resulting from persistent liver damage, is the most significant predictor of morbidity and mortality associated with liver disease, including NAFLD [47,48]. Consequently, recognizing the potential indicators of risk and understanding the significance of a flavonoid treatment that could successfully reduce liver fat accumulation may contribute to the prevention and cure of NAFLD and its progression to a more severe form [49,50].

Researchers have extensively studied the impacts of either pure flavonoids or plant extracts rich in flavonoids on the liver function of NAFLD patients [51,52,53,54,55,56]. A twelve-week treatment with 500 mg of quercetin decreased intrahepatic lipid content from 11.5% ± 6.4% to 9.6% ± 5.8% in 36 patients with confirmed NAFLD [52]. Interestingly, the decreased level of intrahepatic lipid content in females was about twice as large as that in males (3.0% ± 3.7% vs. 1.4% ± 2.5%). In another trial, consumption of a nutraceutical including a Bergamot polyphenol fraction and a flavonoid-rich extract from *Cynara cardunculus* (300 mg/day) for 12 weeks reduced liver fat levels in 45 patients with liver steatosis. Moreover, an improvement in liver elasticity parameters was observed where the controlled attenuation parameter (CAP) score reduction was statistically significant only in individuals over the age of 50 (44% vs. 78% in placebo and nutraceutical) [51]. It should be highlighted that the extracts used in this intervention treatment included such flavonoids as naringenin, luteolin, and apigenin. Similarly, the beneficial effect of epigallocatechin gallate (EGCG) was reported in a study with 20 participants with NAFLD. Supplementation of the diet with 300 mg of EGCG for 24 weeks reduced hepatic fat content compared to the baseline (−11.27 ± 13.85 dB/m) [55]. Furthermore, EGCG from green tea inhibited the expression and activity of dipeptidyl peptidase 4 (DPP4), which is closely linked to the progression of hepatic steatosis and liver damage [55,57].

Notarnicola et al. determined the effects of whole oranges on liver function. The study included a daily intake of 400 g of this fruit in 62 individuals with MASLD [53]. After 4 weeks of orange supplementation, a 30% reduction in liver steatosis was observed. Importantly, the notable reduction in hepatic steatosis prevalence was observed independently of changes in body weight. Nevertheless, no substantial alterations were observed in fibrosis or plasma liver enzymes [53]. Consistent with this, a recent study confirmed that patients with NAFLD who supplemented their diet with isoflavone (100 mg daily) for 12 weeks decreased their CAP score and improved their steatosis severity [58]. Similarly to the Notarnicola et al. results, no significant change in fibrosis grade was observed.

While the evidence consistently shows no effect of flavonoids on fibrosis grade, one study demonstrates their potential beneficial impact on fibrosis. It is interesting to note that a prospective study involving 452 patients with the human patatin-like phospholipase domain-containing three genes (PNPLA3), which is recognized as the key genetic determinant of NAFLD, reported an inverse association between the consumption of isoflavone-rich foods and the risk of severe fibrosis (stage of fibrosis ≥ 2) amongst carriers of the rs738409 G-allele [54]. This suggests that genetic factors may contribute to modulating the effects of flavonoid consumption on fibrosis progression.

Collectively, these studies show that flavonoid-rich interventions, including quercetin, bergamot flavonoids, EGCG, oranges, and isoflavones, may reduce liver fat content and improve steatosis severity in NAFLD individuals. Interestingly, some compounds demonstrated greater efficacy in specific subgroups, such as older patients or females.

### 5.2. Flavonoids and Their Impact on Liver Biochemical Parameters in NAFLD Patients

It is recognized that elevated liver enzymes are the predominant condition in patients with NAFLD [59,60]. Therefore, reducing the ALT and AST levels may indicate an improvement in the liver function and liver cell damage. Numerous meta-analyses of randomized and prospective studies on the effects of flavonoids on liver enzymes in NAFLD patients suggested a relationship between the various flavonoid interventions and a significant reduction in the liver enzymes, such as ALT and AST levels [10,24,61,62,63,64,65]. A decrease in the liver enzymes was noted in studies with silymarin, naringenin, and dihydromyricetin.

The substance most frequently investigated in clinical studies and meta-analyses is silymarin. Although silymarin is not a classical flavonoid, it is often considered a bioactive flavonoid-related compound due to its antioxidant, anti-inflammatory, and hepatoprotective properties. A recent meta-analysis evaluated the relationship between the administration of silymarin and NAFLD progression [62]. This study involved 26 randomized controlled trials, including 2375 individuals, and observed that treatment with silymarin was associated with a reduction in ALT (SMD = −12.39) and AST (SMD = −10.97) in NAFLD patients. Furthermore, silymarin effectively reduced hepatic fat accumulation and improved hepatic steatosis [62]. Therefore, the protective effects of silymarin in NAFLD could be mediated by an attenuation of liver damage and improvement in liver histology.

Another meta-analysis of prospective cohort studies, which included 12 trials with 418 NAFLD men and women, reported a significant reduction in liver enzyme levels after flavonoid intake [10]. Li et al. found that flavonoids, including hesperidin, silybum, anthocyanin, genistein, and dihydromyricetin, decreased ALT (SMD = −3.59, *p* = 0.034), AST (SMD = −4.47, *p* = 0.001), and GGT (SMD = −8.70, *p* = 0.000). Similarly, two independent meta-analyses investigated different flavonoid-based interventions for NAFLD. One study examined the effects of traditional Chinese medicine together with silibinin, while the other analyzed various natural compounds, including silymarin, artichoke leaf extract, berberine, catechins, and naringenin. Despite focusing on different interventions, both meta-analyses confirmed that flavonoid-rich interventions significantly reduced ALT and AST in NAFLD patients [63,65]. Given these data, it is tempting to speculate that various flavonoid-based interventions effectively lower liver enzyme levels in patients with NAFLD.

### 5.3. Flavonoids, Metabolic Status, and Body Weight in NAFLD

The evidence over the past decades and current research have shown the potential benefits of flavonoids in improving the metabolic status of individuals with NAFLD [66,67]. The prevalence of cardiovascular diseases (CVDs) is increasing among patients with non-alcoholic fatty liver disease, and patients diagnosed with NAFLD have an elevated mortality rate from CVD events [68,69,70]. Therefore, diminished CVD morbidity and mortality is a key target for intervention in NFLD treatment.

The effect of flavonoids on lipids, insulin resistance, body weight, and other metabolic markers associated with NAFLD progression was widely investigated. Neshatbini et al. evaluated the effect of soy isoflavones. They showed that compared to the baseline, isoflavone treatment (100 mg/day for 12 weeks) improved serum TGs (−43.68%), LDL-c (−16.28%), and TC (−13.52%) in 50 NAFLD patients [56]. Nevertheless, no beneficial effects were observed on high-density lipoprotein cholesterol, blood pressure, or glycemic parameters. Two publications based on the same patient cohort reported the beneficial effects of naringenin on lipids [71,72]. A four-week diet supplementation with 200 mg/day of naringenin reduced TGs, TC, and LDL-C in 44 individuals with NAFLD. These results were accompanied by an increase in HDL-c [72]. In a further analysis, Naeini et al. reported improvements in cardiovascular risk factors, including a reduction in the atherogenic index of plasma and a decrease in Body Mass Index (BMI) [71]. Furthermore, a combination therapy of flaxseed and hesperidin (30 g + 1000 mg/day) for 12 weeks decreased plasma triglycerides (MD = −45.19 mg/dL), total cholesterol (MD = −26.53 mg/dL), fasting glucose (MD = −6.12 mg/dL), and indices of HOMA-IR (MD = −1.51) and insulin (MD = −3.17 mU/L) in NAFLD patients [73]. This suggests the positive role of this combined treatment in metabolic regulation.

Elevated uric acid levels are commonly related with metabolic dysfunction in NAFLD patients [74,75]. Hyperuricemia is linked to insulin resistance, oxidative stress, and inflammation, all of which contribute to the progression of NAFLD [76]. Ferro et al. investigated the effects of flavonoids from bergamot and *Cynara cardunculus* extract (300 mg/day) on the metabolic parameters in individuals with hepatic steatosis [77]. After six weeks of intervention, a serum uric acid (SUA) change was observed, especially in participants with severe hepatic steatosis. Furthermore, the participants with the highest baseline serum uric acid (>5.4 mg/dL) experienced a more significant decrease than those with the lowest baseline SUA (−7.8% vs. +4.9%).

Meta-analyses of clinical randomized trials further support the beneficial role of flavonoids in their potential lipid-modulating effects among patients with NAFLD [10,24,62,63,65]. Yang et al. analyzed the results from 46 studies involving participants with NAFLD and assessed the effectiveness of dietary flavonoids in the treatment of NAFLD [24]. The analysis showed that the consumption of naringenin, hesperidin, and catechin by NAFLD patients had a beneficial effect on lipid profiles. Among these substances, naringenin exhibited beneficial effects on most components of the lipid profiles. Consistent with these findings, another meta-analysis showed that artichoke leaf extract and naringenin significantly improved lipid profiles in NAFLD [63]. While these findings consistently highlight the beneficial role of various flavonoids in modulating lipid profiles in NAFLD, some heterogeneity was noted across the studies due to variations in flavonoid type, dosage, intervention duration, and patient ethnicity, and therefore, further research is warranted.

### 5.4. Flavonoids and Anti-Inflammatory Effects in NAFLD

Inflammation plays a crucial role in the progression of non-alcoholic fatty liver disease. It is recognized that a chronic low-grade tissue inflammatory state increases the risk of progressive fibrosis, cirrhosis, liver failure, and hepatocellular carcinoma in NFLD patients [78]. Therefore, preventing chronic inflammation may delay or prevent inflammation-associated liver damage.

Several studies have shown that flavonoid supplementation reduces inflammatory markers in NAFLD patients [3,79]. Grape seed extract (GSE), which is rich in proanthocyanidins, was shown to modulate inflammation in NAFLD individuals. Compared with the baseline, two months of administration of GSE (250 mg of proanthocyanidins/daily) significantly reduced IL-6 (27.3 ± 1.87 pg/mL vs. 25.2 ± 3.60 pg/mL) and malondialdehyde (17.0 ± 1.95 μM vs. 15.5 ± 2.55 μM) in 25 NAFLD patients [3]. GSE treatment also increased the levels of antioxidant enzymes, such as SOD (48.3 ± 7.95 U/m vs. 55.0 ± 6.49 U/m), and improved the total antioxidant capacity (TAC) (0.48 ± 0.08 mM vs. 0.56 ± 0.14 mM). In addition to the above studies, which focus on proanthocyanidin mixtures, one study specifically investigates the effects of pure anthocyanin intake. Anthocyanins (160 mg/day for 12 weeks) were shown to decrease pro-inflammatory cytokines, including caspase-1, IL-1β, and IL-18, in NAFLD patients [79]. Anthocyanin also downregulated the NLRP3 inflammasome expression associated with the pathogenesis of NAFLD.

A recent meta-analysis including 12 randomized controlled trials also supports the beneficial effect of dietary flavonoids, including hesperidin, silybum, anthocyanin, genistein, and dihydromyricetin, on inflammation [10]. Flavonoids were shown to reduce inflammatory markers, including tumor necrosis factor-alpha (TNF-α) (MD= −0.88, *p* = 0.000) and nuclear factor kappa-light-chain-enhancer of activated B cells (NF-κB) (MD= −1.62, *p* = 0.001). These findings suggest that flavonoids may exert hepatoprotective effects by modulating inflammatory pathways and oxidative stress. However, the results from the obtained data reflected the patients’ systemic inflammation status rather than their hepatic inflammation status.

## 6. Summary

Prospective, clinical, animal, and in vitro studies support the beneficial role of many flavonoids, including quercetin, epigallocatechin gallate, naringenin, and isoflavones, in the management of different stages of NAFLD. The results of randomized controlled trials on the beneficial effects of flavonoids among non-alcoholic fatty liver disease patients are synthesized in this narrative review. Over the past five years, at least 13 clinical studies have investigated the correlation between flavonoid consumption and the clinical and biochemical parameters associated with NAFLD.

The data from the clinical trials suggest that flavonoid interventions effectively reduce liver fat accumulation and improve metabolic parameters in NAFLD patients. However, the long-term effectiveness of flavonoid supplementation remains uncertain, particularly regarding fibrosis and cirrhosis progression. Though flavonoid supplements have been linked to beneficial changes in liver enzymes, lipid metabolism, and liver steatosis, their impact on fibrosis remains unclear. Except for a noted inverse connection between isoflavone intake and fibrosis advancement in genetically predisposed individuals, most studies show no appreciable difference in fibrosis severity. The temporary character of the improvements in hepatic steatosis without corresponding reductions in fibrosis raises concerns about the sustainability of flavonoid-related benefits. These results emphasize the need for more research to understand the long-term effects of flavonoid intake on liver health beyond steatosis reduction.

Additionally, the efficacy of flavonoids seems to vary based on variables such as age, sex, and metabolic status, with some flavonoids demonstrating enhanced advantages in particular populations, including older adults and females. However, the fundamental mechanisms behind these disparities remain unexamined in the current review. It seems that a more detailed analysis of gene–diet interactions and individualized patient responses could improve treatment personalization and optimize flavonoid-based therapies.

Flavonoid intake among patients with NAFLD seems to be both safe and efficacious. Nonetheless, the limitations of the conducted investigations were also noted. The primary limitations of the current research include the small sample sizes, the short duration of the presented investigations (ranging mostly from 4 to 12 weeks), and the heterogeneity of the flavonoid types and dosages used. Moreover, most randomized controlled trials that were used in this review failed to account for flavonoid consumption from dietary sources, which may influence the final results. Additionally, the use of different methods to assess liver steatosis across the RCTs might not be suitable to evaluate the steatosis stages in detail. Moreover, potential biases—such as the homogeneity of the study populations, the limited follow-up periods, and the lack of standardized diagnostic criteria—must be carefully considered when interpreting the findings. Other potential biases include the lack of ethnic diversity across the intervention groups and the insufficient assessment of comorbidities that may affect the outcomes. Additionally, the diversity of the flavonoid types examined complicates the establishment of standardized treatment protocols. These differences in study design highlight the need for standardized methodologies in future research.

In conclusion, the studies of this review provide evidence that a flavonoid-rich diet can help manage NAFLD. To advance the flavonoid research in NAFLD management, large-scale, long-term clinical trials with well-defined intervention protocols and standardized assessment methods are necessary. Further studies should focus on clear guidelines for flavonoid-based therapies and explore their role in preventing fibrosis and cirrhosis. The data from this review would help medical practitioners comprehend the potential benefits and limitations of flavonoids in the management of non-alcoholic fatty liver disorders.

## Figures and Tables

**Table 1 nutrients-17-00956-t001:** Content of flavonoids in chosen foodstuffs (mg/100 g foodstuff); authors’ selection based on [22,23].

Flavonoids (mg/100 g)					
Fruits		Vegetables		Spices and Herbs	
Elderberries	518.05	Cabbage, red	210.16	Parsley, dried	4854.49
Chokeberries	368.66	Dock	102.20	Oregano, Mexican, dried	1545.79
Currants, black	170.41	Kale	92.98	Capers, raw	493.03
Blueberries	180.82	Fennel	84.50	Parsley, fresh	233.16
Blackberries	147.63	Radishes	63.99	Peppermint, fresh	60.48
Cranberries	99.22	Onions, red	56.61	Thyme, fresh	47.75
Currants, red	79.49	Arugula	47.11	Nuts	
Kumquats	79.26	Chard	25.60	Pecans	34.01
Raspberries	55.57	Celery hearts, green	22.60	Pistachios	14.37
Grapefruits	55.40	Artichokes	22.28	Hazelnuts	11.96
Lemons	53.38	Peppers, hot chili	21.17	Almonds	11.00
Grapes, red	52.42	Broadbeans, cooked	20.63	Beverages	
Limes	46.80	Chives	17.11	Black tea, brewed	118.35
Oranges	43.49	Asparagus, cooked	15.16	Black currant juice	78.04
Cherries	40.00	Cress, fresh	14.00	White tea, brewed	74.60
Strawberries	33.52	Chicory, green	11.79	Green tea, brewed	53.84
Plums	14.42	Lettuce, red	11.72	Oolong tea, brewed	52.37
Apples	13.73	Spinach	11.44	Wine, table red	34.49
Apricots	10.67	Endive	10.10	Orange juice	24.13
Pears	8.01	Brussels sprouts, cooked	7.68	Pink grapefruit juice	17.97
Bananas	6.21	Peppers, green	6.98	Beer	1.39
Legumes		Sweets		Cereals and grains	
Cowpeas, black	277.41	Dark chocolate	108.60	Wheat, purple	25.85
Beans, black	28.00	Cocoa, dry powder	106.68	Buckwheat	15.38
Beans, kidney, red	10.87	Milk chocolate	15.04	Sorghum, red	8.43

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
