# Peer review of "Clinical Insights into Non-Alcoholic Fatty Liver Disease and the Therapeutic Potential of Flavonoids: An Update"

_nutrients, 2025, doi:10.3390/nu17060956_

Round 1
Reviewer 1 Report
Comments and Suggestions for Authors
This review is based on the therapeutic potential of flavonoids for managing non-alcoholic fatty liver disease (NAFLD).
While the review includes 20 studies, with 13 RCTs and seven meta-analyses, it may not provide a comprehensive picture of the current state of research. The selection criteria could potentially overlook relevant studies, leading to a biased understanding of flavonoids’ efficacy.
The review emphasizes the reduction of hepatic steatosis and improvement in liver enzyme profiles, but it downplays the importance of long-term outcomes, particularly regarding fibrosis and cirrhosis progression. The transient nature of improvements in liver fat content without significant fibrosis reduction raises questions about the long-term effectiveness of flavonoid interventions.
While certain flavonoids show superior efficacy in subgroups like older adults and females, the review does not explore the underlying mechanisms for these differences. A more nuanced understanding of how factors like age, sex, and metabolic status influence flavonoid effects could enhance treatment personalization.
The call for extensive, controlled clinical trials to create uniform treatment methods is valid, but the review does not specify what these methods might entail. Without clear guidelines or protocols, the path forward for research and clinical application remains vague.
Author Response
Review 1.
This review is based on the therapeutic potential of flavonoids for managing non-alcoholic fatty liver disease (NAFLD).
Comments 1: While the review includes 20 studies, with 13 RCTs and seven meta-analyses, it may not provide a comprehensive picture of the current state of research. The selection criteria could potentially overlook relevant studies, leading to a biased understanding of flavonoids’ efficacy.
Response 1: Thank you for pointing this out. I agree with these comments. In the presented review, I focused on studies published in the last five years, using appropriate queries in PubMed. To clarify this, I refined the methodology section to provide a more detailed explanation of the selection criteria. Additionally, potential biases related to study selection have been explicitly addressed in the summary section to ensure a more transparent discussion of the limitations.
Comments 2: The review emphasizes the reduction of hepatic steatosis and improvement in liver enzyme profiles, but it downplays the importance of long-term outcomes, particularly regarding fibrosis and cirrhosis progression. The transient nature of improvements in liver fat content without significant fibrosis reduction raises questions about the long-term effectiveness of flavonoid interventions.
Response 2:Thank you for this valuable comment. Unfortunately, no studies on the long-term effects have been published in recent years. This is an interesting aspect, and I have incorporated this observation into the summary section. ”Except for a noted inverse connection between isoflavone intake and fibrosis advancement in genetically predisposed individuals, most studies show no appreciable difference in fibrosis severity. The temporary character of improvements in hepatic steatosis without corresponding reductions in fibrosis raises concerns about the sustainability of flavonoid-related benefits. This result emphasizes the need for more research to understand the long-term effects of flavonoid intake on liver health beyond steatosis reduction”
Comments 3: While certain flavonoids show superior efficacy in subgroups like older adults and females, the review does not explore the underlying mechanisms for these differences. A more nuanced understanding of how factors like age, sex, and metabolic status influence flavonoid effects could enhance treatment personalization.
Response 3: Thank you for pointing this out. I agree with these comments. I incorporated this point into the summary section. I believe that understanding these mechanisms is essential for guiding future research. I hope that my work will help other researchers design studies that clarify why these effects are observed in certain groups but not in others.
Comments 4: The call for extensive, controlled clinical trials to create uniform treatment methods is valid, but the review does not specify what these methods might entail. Without clear guidelines or protocols, the path forward for research and clinical application remains vague.
Response 4: Thank you for this valuable comment. I agree with these comments. Since flavonoids represent a large group of compounds, I believe that future research should focus primarily on individual substances and their specific effects on NAFLD. Additionally, distinguishing the severity of NAFLD through appropriate imaging diagnostics is crucial, as this is often a major source of bias in many studies. I have incorporated these conclusions into the summary section. “Additionally, the efficacy of flavonoids seems to vary based on variables such as age, sex, and metabolic status, with some flavonoids demonstrating enhanced advantages in particular populations, including older adults and females. However, the fundamental mechanisms behind these disparities remain unexamined in the current review. It seems that a more detailed analysis of gene-diet interactions and individualized patient responses could improve treatment personalization and optimize flavonoid-based therapies.” “. Moreover, potential biases—such as the homogeneity of study populations, limited follow-up periods, and the lack of standardized diagnostic criteria—must be carefully considered when interpreting findings. Other potential biases include the lack of ethnic diversity across intervention groups, and insufficient assessment of comorbidities, which may affect outcomes. Additionally, the diversity of flavonoid types examined complicates the establishment of standardized treatment protocols. These differences in study design highlight the need for standardized methodologies in future research.”
Reviewer 2 Report
Comments and Suggestions for Authors
Dear corrisponding Author, thank you for submitting your work to the journal Nutrients and congratulations on your research.
1) Brief Summary
This manuscript presents a narrative review on the therapeutic potential of flavonoids in NAFLD. The author selected 20 recent studies (13 RCTs and 7 meta-analyses) examining the effects of flavonoids on NAFLD parameters, describing dietary sources, mechanisms of action, and clinical results showing reduction in hepatic fat, improvement in liver enzymes, and positive effects on lipid metabolism and inflammation.
2) General Comments
- The introduction needs a more in-depth background on the pathogenesis of NAFLD to better understand the rationale for using flavonoids.
- The methodology of the literature search is incomplete: precise dates, clear exclusion criteria, and description of the selection process are missing.
- The results could benefit from a more systematic structure, categorizing by flavonoid type, dosage, and outcome.
- Potential biases of the included studies are not adequately discussed!
- Practical implications for clinical practice and the "Author Contributions" section are missing. Why?
3) Specific Comments
- Line 43-44: Better contextualize the statement about limited treatment options. And check the punctuation because some sentences are strangely interrupted, for example "Despite the prevalence of MAFLD and NASH, effective treatment options remain limited. (???) Lifestyle interventions..."
- Line 53-55: Critically analyze the relevance of the ClinicalTrials.gov studies cited.
- Line 59-71: Specify the selection method and data extraction. For example, in line 76 you wrote "author meticulously evaluated 20 studies", what does this mean? By what criteria? How many initial studies were there after the selection you indicated? It's all very summary
- Lines 101-111: Provide concrete data on flavonoid safety. For example, "high dosages" seems generic for such a relevant statement
- Lines 357-366: Elaborate on methodological limitations. Include an extensive limitations section because the review appears limited in terms of study selection with a high risk of bias that should be well clarified along with other criticalities.
- The Author Contributions section does not exist. Even if it's only one author, I believe it should still be included
The work is interesting and well-structured; I believe it can be corrected in a some points and then resubmitted. I look forward to reading a final version to give my definitive evaluation
Author Response
Review 2
1) Brief Summary
This manuscript presents a narrative review on the therapeutic potential of flavonoids in NAFLD. The author selected 20 recent studies (13 RCTs and 7 meta-analyses) examining the effects of flavonoids on NAFLD parameters, describing dietary sources, mechanisms of action, and clinical results showing reduction in hepatic fat, improvement in liver enzymes, and positive effects on lipid metabolism and inflammation.
2) General Comments
Comments 1: The introduction needs a more in-depth background on the pathogenesis of NAFLD to better understand the rationale for using flavonoids.
Response 1: Thank you for pointing this out. I agree with these suggestions. The introduction has been revised to include a more in-depth background on the pathogenesis of NAFLD, which will provide a clearer understanding of the rationale for using flavonoids. “NAFLD is mainly caused by overnutrition, which results in the accumulation of visceral fat. In this context, the infiltration of macrophages into the visceral adipose tissue compartment induces a proinflammatory state that leads to insulin resistance. Thus, it leads to an increased influx of free fatty acids into the liver, enhanced de-novo lipogenesis, and impaired fatty acid oxidation. At the same time, dysregulation of lipid metabolism results in the accumulation of lipotoxic lipids, which induce cellular stress, which activates inflammatory pathways and causes hepatocyte damage. The pathogenic mechanisms of NAFLD are also affected by several metabolic, genetic, and microbiome-related variables. However, they remain not fully understood”
Comments 2:The methodology of the literature search is incomplete: precise dates, clear exclusion criteria, and description of the selection process are missing.
Response 2: Thank you for this valuable comment. I agree with this comment. In the methodology, I took into account queries from PubMed, focusing on studies published in the last five years, specifically RCTs and meta-analyses. Any studies that were included by chance were excluded. The methodology section has been revised and clarified to better reflect these details. Thank you once again for your thoughtful feedback.
“The search was performed on January 20, 2025, and included studies published between January 1, 2020, and January 20, 2025. For study selection, the author applied the following inclusion criteria: (1) randomized controlled trials (RCTs); (2) clinical trials; (3) meta-analyses; (4) studies published in English; and (5) studies published in the last 5 years. The exclusion criteria were: (1) studies not available in English, (2) case reports, (3) narrative reviews, and (4) studies focusing on nonclinical (in vitro or animal model) outcomes. After screening titles and abstracts, studies that did not meet the inclusion criteria, or contained misleading data were excluded, resulting in 20 eligible studies (13 RCTs and 7 meta-analyses). The selection process involved assessing study relevance based on predefined inclusion and exclusion criteria, followed by full-text screening. Data extraction was performed using EndNote X7, focusing on study design, population characteristics, intervention details, flavonoid type, dosage, duration, and primary clinical outcomes (Supplementary Table 1). Preclinical data were not limited regardless of their publication by publication date. However, the author has included a review of the most recent studies available up to the current date that the readers may not be familiar with.”
Comments 3: The results could benefit from a more systematic structure, categorizing by flavonoid type, dosage, and outcome.
Response 3: Thank you for this valuable comment. I completely agree with your suggestion. After completing this review, I have concluded that studies focusing on specific flavonoid compounds would provide more detailed and refined results. Flavonoids represent a very large group of substances, but based on this review, it is evident which compounds are most commonly used in interventions, and these should be the focus of future research. In my opinion, this provides a valuable resource for clinicians designing studies, helping them identify the most promising substances for further investigation.
Comments 4:Potential biases of the included studies are not adequately discussed!
Response 4: Thank you for pointing this out. I agree with these suggestions. The Summary section has been expanded and updated to include a discussion of potential biases in the included studies. Additionally, the main limitations of each study are listed in the supplementary materials.
“The temporary character of improvements in hepatic steatosis without corresponding reductions in fibrosis raises concerns about the sustainability of flavonoid-related benefits. This result emphasizes the need for more research to understand the long-term effects of flavonoid intake on liver health beyond steatosis reduction.
Additionally, the efficacy of flavonoids seems to vary based on variables such as age, sex, and metabolic status, with some flavonoids demonstrating enhanced advantages in particular populations, including older adults and females. However, the fundamental mechanisms behind these disparities remain unexamined in the current review. It seems that a more detailed analysis of gene-diet interactions and individualized patient responses could improve treatment personalization and optimize flavonoid-based therapies.
Flavonoid intake among patients with NAFLD seems to be both safe and efficacious. Nonetheless, the limitations of the conducted investigations were also noted. The primary limitations of the current research include the small sample sizes, the short duration of the presented investigation (ranging mostly from 4 to 12 weeks), and the heterogeneity of flavonoid types and dosages used. Moreover, most randomized controlled trials that were used in this review failed to account for flavonoid consumption from dietary sources, which may influence the final results. Additionally, the use of different methods to assess liver steatosis across RCTs might not be proper to evaluate steatosis stages in detail. Moreover, potential biases—such as the homogeneity of study populations, limited follow-up periods, and the lack of standardized diagnostic criteria—must be carefully considered when interpreting findings. Other potential biases include the lack of ethnic diversity across intervention groups, and insufficient assessment of comorbidities, which may affect outcomes. Additionally, the diversity of flavonoid types examined complicates the establishment of standardized treatment protocols. These differences in study design highlight the need for standardized methodologies in future research.”
Comments 5:Practical implications for clinical practice and the "Author Contributions" section are missing. Why?
Response 5: Thank you for pointing this out. I agree with these suggestions. I added Author Contributions section.
3) Specific Comments
- Line 43-44: Better contextualize the statement about limited treatment options. And check the punctuation because some sentences are strangely interrupted, for example "Despite the prevalence of MAFLD and NASH, effective treatment options remain limited.(???) Lifestyle interventions..."
- Line 53-55: Critically analyze the relevance of the ClinicalTrials.gov studies cited.
- Line 59-71: Specify the selection method and data extraction. For example, in line 76 you wrote "author meticulously evaluated 20 studies", what does this mean? By what criteria? How many initial studies were there after the selection you indicated? It's all very summary
- Lines 101-111: Provide concrete data on flavonoid safety. For example, "high dosages" seems generic for such a relevant statement
- Lines 357-366: Elaborate on methodological limitations. Include an extensive limitations section because the review appears limited in terms of study selection with a high risk of bias that should be well clarified along with other criticalities.
- The Author Contributions section does not exist. Even if it's only one author, I believe it should still be included
Response to Specific Comments:
I agree with the specific comments, and the changes have been made to the text. I would like to highlight that I used the example of clinical trial registration in ClinicalTrials.gov to emphasize that flavonoids are currently being used in clinical research. To clarify this point, I have added a corresponding sentence to the Introduction. “Nevertheless, the rising interest in the potential therapeutic applications of flavonoids is evidenced by the increasing number of clinical research assessing their efficacy and safety across diverse health conditions. In a query conducted in the ClinicalTrials.gov database in February 2025, the keyword “flavonoid” yielded 179 clinical studies in which these compounds were investigated.”
The work is interesting and well-structured; I believe it can be corrected in a some points and then resubmitted. I look forward to reading a final version to give my definitive evaluation
Thank you very much for your review. Your feedback was extremely helpful in improving the manuscript, and I truly appreciate the time and effort you put into it. Your insightful comments have guided me in refining the work, and I look forward to sharing the final version with you. I sincerely appreciate your support.
Reviewer 3 Report
Comments and Suggestions for Authors
In manuscript nutrients-3517368, Kozłowska discusses the potential clinical applications of flavonoids in MAFD. The topic of this review is interesting and fits well the scope of Nutrients. However, the reviewer feels it needs extensive amendments before it can be accepted.
(1) The health promoting activities of flavonoids are well-known. However, the potency is quite low and high dose is required. So, the authors should discuss this point.
(2) The polyphenol type of flavonoids are metabolically instable and the oral bioavailability is generally poor. The authors must discuss the impact of such issue on their clinical translation.
(3) The molecular mechanism of action should be discussed.
(4) As a review, graphical presentation is important. The author should improve the presentation and make this manuscript more attractive. A colorful figure will be very helpful.
Author Response
Review 3.
Comments 1: The health promoting activities of flavonoids are well-known. However, the potency is quite low and high dose is required. So, the authors should discuss this point.
Comments 2: The polyphenol type of flavonoids are metabolically instable and the oral bioavailability is generally poor. The authors must discuss the impact of such issue on their clinical translation.
Response 1+ 2: Thank you for pointing this out. I agree with these suggestions. The section Introduction has been updated to include this information. I have addressed this issue by expanding the Introduction on the metabolic instability and low oral bioavailability of flavonoids. I hope now it is more readable and highlights flavonoid implications for clinical translation. Thank you.
“These substances, which are recognized for their antioxidant, anti-inflammatory, and lipid-lowering properties, have been proposed to provide substantial metabolic health benefits to patients with hepatic steatosis. However, despite their promising biological activities, flavonoids frequently demonstrate relatively low effectiveness, which frequently requires high doses of these substances to obtain therapeutic effects. Additionally, the poor bioavailability of many flavonoids presents significant challenges for their clinical translation. Strategies such as structural modifications, nanoformulations, and co-administration with enhancers are being explored to overcome these limitations”
Comments 3: The molecular mechanism of action should be discussed.
Response 3: Thank you very much for this valuable comment. The molecular mechanism of action has been expanded in the subsection Flavonoids and Their Action in NAFLD, where I have described the specific pathways related to NAFLD. Additionally, table 2 is included to enhance clarity. I hope this provides sufficient details, as I have already exceeded the character limit for this manuscript and would like to focus on clinical trial results. I truly appreciate this suggestion and your thoughtful feedback.
Comments 4: As a review, graphical presentation is important. The author should improve the presentation and make this manuscript more attractive. A colorful figure will be very helpful.
Response 4: Thank you for this suggestion. I completely agree that graphical presentation is important. This manuscript is already quite extensive, with multiple tables and a comprehensive supplementary section, so unfortunately, I am unable to add more figures at this stage. However, I truly appreciate this insight and will certainly take it into account in future work.
Round 2
Reviewer 2 Report
Comments and Suggestions for Authors
I have carefully reviewed the changes made by the authors to my previous comments. I believe the paper is now more coherent and can be submitted for publication in its current form.
Reviewer 3 Report
Comments and Suggestions for Authors
The manuscript has been improved and the reviewer has no objection to pass it.